# The Effects of Value Conflicts on Stress in Chinese College Students: A Moderated Mediation Model

**DOI:** 10.3390/bs15020104

**Published:** 2025-01-21

**Authors:** Xiaoxiao Ren, Hao Xu, Tong Yue, Tong Wang

**Affiliations:** 1School of Psychology, Shaanxi Normal University, Xi’an 710062, China; 2Preschool Education Department, Chongqing Preschool Education College, Chongqing 404047, China; 3Faculty of Psychology, Southwest University, Chongqing 400715, China; 4Shaanxi Teacher Development Institute, Shaanxi Normal University, Xi’an 710062, China

**Keywords:** value conflicts, self-concept clarity, stress, mental health, Chinese college students, Schwartz’s value theory

## Abstract

Limited research has explored the connection between stress and value conflicts, particularly the influence of self-construal and self-concept clarity. This study surveyed a sample of 752 Chinese college students using the Depression–Anxiety-Stress Scale, Self-Construal Scale, Self-Concept Clarity Scale, and Portrait Values Questionnaire. The findings demonstrated that stress levels among participants were significantly associated with conflicts between self-transcendence and self-enhancement values, but no significant relationship was observed with openness vs. conservation conflicts. Mediation analysis revealed that self-concept clarity partially mediated this relationship. Additionally, moderated mediation analysis showed that the association between value conflict and self-concept clarity was stronger in students with high levels of independent self-construal. These results offer a deeper understanding of how value conflicts contribute to stress, highlighting potential pathways for targeted mental health interventions. Future studies should address the limitations of the current research and explore these relationships in more diverse contexts.

## 1. Introduction

[33] ([33]) defines values as “concepts or beliefs related to desirable end-states or behaviors that transcend specific situations and serve as guiding principles in life”. The value model comprises four primary dimensions: self-enhancement, self-transcendence, openness to change, and conservation. Within this model, two core value conflicts exist. The first conflict is between self-transcendence and self-enhancement: self-transcendence emphasizes altruism, compassion, and concern for others, whereas self-enhancement prioritizes personal achievement, status, and power. The second conflict is between conservation and openness to change: conservation tends to emphasize stability and tradition, while openness to change advocates for innovation and exploration ([33], [34]). Research has shown that adherence to consistent values is closely associated with higher well-being ([32]). However, when individuals simultaneously pursue conflicting values, it typically results in significant psychological stress. Value conflicts often lead to approach-avoidance conflicts, which generate internal tension and exacerbate stress. For instance, studies have found that individuals pursuing both materialism and environmental goals—representing a conflict between self-enhancement and self-transcendence—experience substantial psychological distress ([14]). Similarly, individuals who strongly identify with collectivist values (such as religious and familial values) are more likely to experience higher levels of psychological tension or stress when they also endorse materialistic values ([3]). Considering the prevalence of value conflicts across different cultures, this study proposes Hypothesis 1: Among Chinese university students living in an era of diverse coexisting values, individuals who pursue opposing values (e.g., simultaneously valuing self-enhancement and self-transcendence or conservation and openness to change) will report higher levels of psychological stress.

Self-concept clarity (SCC) refers to the extent to which individuals clearly and consistently define their self-concept ([5]). High SCC contributes to stable self-perception and effective goal alignment, whereas low SCC manifests as self-doubt and uncertainty, which can interfere with decision-making and exacerbate stress ([25]). Existing research indicates that, on one hand, value conflicts may compel individuals to prioritize conflicting goals, thereby weakening their ability to maintain a consistent sense of self. This erosion of self-concept is associated with increased anxiety and stress, as low SCC activates the brain’s behavioral inhibition system ([7]; [18]). On the other hand, for university students undergoing educational transitions, emotional regulation abilities can help maintain high self-concept clarity, thereby enhancing psychological resilience and mitigating the negative impacts of stress arising from value conflicts ([27]). Furthermore, it can be inferred that in uncertain situations involving different value conflicts, individuals may experience blurred self-cognition, threatening the integrity of their self-concept, leading to emotional instability and increased stress levels. Accordingly, this study proposes Hypothesis 2: Conflicting values contribute to stress by reducing self-concept clarity, which acts as a mediating factor.

Additionally, research has found that issues with SCC are more prominent in Western cultures, where internal consistency and stability are emphasized, and individuals are expected to maintain a clear sense of identity across different contexts ([5]). In contrast, individuals with an interdependent self-construal typically engage in dialectical thinking, resulting in a less consistent self-concept, which may allow them to better tolerate value conflicts ([26]; [29]). Indeed, studies have shown that compared to Chinese university students with independent self-construal, those with interdependent self-construal exhibit greater tolerance for conflicts when pursuing opposing values, resulting in smaller negative impacts on psychological health. However, it is noteworthy that with globalization and China’s reform and opening-up, an increasing number of Chinese youth are adopting independent self-construals rather than the traditionally interdependent self-construals ([21]; [43]). This provides a valuable socio-cultural context for exploring how value conflicts and self-concept clarity influence stress and psychological health. Considering that independent self-construal emphasizes autonomy and personal goals, it may exacerbate the impact of value conflicts on SCC by forcing individuals to resolve internal inconsistencies ([12]). Therefore, it is necessary to investigate how independent self-construal moderates the relationship between value conflicts and SCC within the Chinese cultural context to further reveal how socio-cultural variables shape the relationship between value conflicts, SCC, and stress. Accordingly, this study proposes Hypothesis 3: Independent self-construal moderates the relationship between value conflicts and SCC. Specifically, Chinese students with a higher degree of independent self-construal will exhibit a stronger relationship between value conflicts and SCC compared to those with lower levels.

Based on the aforementioned hypotheses, this study proposes a moderated mediation model (see Figure 1) aimed at investigating the impact of value conflicts on perceived stress among Chinese university students, the mediating role of SCC, and the moderating effect of independent self-construal. On one hand, by elucidating the mediating role of SCC, the findings of this study can advance the understanding of how value conflicts influence mental health. On the other hand, by examining the moderating effect of independent self-construal, this study reveals the complex interplay between socio-cultural changes and the development of contemporary Chinese youths’ values.

## 2. Methods

### 2.1. Design

This study employed a cross-sectional survey design to investigate the relationships between value conflict, SCC, stress, and independent self-construal among Chinese college students. Ethical approval for the study was obtained from the Ethics Committee of Southwest University (IRB NO. H24040), and all procedures adhered to the provisions of the Declaration of Helsinki. Participants were informed of the study’s purpose and procedures before providing written informed consent.

### 2.2. Setting

The study was conducted across three regions in China: Chongqing, Shandong, and Tianjin. Data collection involved distributing self-report questionnaires to students from multiple universities in these regions. The survey was administered both online and offline, ensuring accessibility and convenience for participants.

To ensure consistency in the data collection process across the different universities and regions, all participants were provided with identical instructions on how to complete the survey. In addition, researchers were trained to administer the survey uniformly, regardless of the mode of delivery (online or offline). This standardized approach aimed to minimize any potential regional or institutional biases that could influence the responses.

### 2.3. Participants

This study was designed to test a moderated mediation analysis model. To ensure a sufficient sample size for detecting the expected effects, a priori power analysis was conducted using G*Power 3.1. Assuming a medium effect size (f^2^ = 0.15), a significance level of α = 0.05, statistical power of 0.80, and 7 predictor variables in the regression model, the analysis indicated that a sample size of 106 participants would be required to detect the contribution of the moderating variables and interaction terms.

A total of 776 college students were initially recruited, with a mean age of 18.68 years (SD = 2.23), meeting the complexity requirements of the model. After excluding incomplete responses and questionnaires with overly regular or chaotic answers (24 in total), 752 valid responses were retained, resulting in an effective response rate of 96.91%.

For data analysis, a listwise deletion method was employed to handle the small proportion of missing values (<5%), ensuring the integrity of the model and data quality. Among the participants, 210 were male (27.93%) and 542 were female (72.07%). Students from all academic years were included: 306 freshmen, 157 sophomores, 188 juniors, and 101 seniors. This diverse sample ensured representation across various demographic and academic characteristics.

### 2.4. Outcome Measures

The study utilized several standardized and validated instruments to measure the key variables of personal values, value conflict, self-concept clarity, stress, and self-construal.

Personal values were assessed using the Chinese version of the Portrait Values Questionnaire (PVQ-21), which was adapted by [15] ([15]) from the original scale developed by [35] ([35]). PVQ-21 consists of 21 portrait descriptions, each representing a value domain by subtly reflecting an individual’s goals or aspirations. For example, one portrait states, “For him, creativity and ingenuity are very important. His style of doing things is original”, highlighting the value of self-direction. Respondents rated how similar each description was to themselves on a 6-point scale, ranging from 1 (“very similar”) to 6 (“not at all”). There were no items requiring reverse scoring. To account for individual differences, the relative importance of each value was determined by centering the responses around the participants’ average score. In this study, the internal consistency of the four subscales ranged from α = 0.67 (self-enhancement) to α = 0.74 (openness).

The magnitude of value conflict was calculated using a formula adapted from previous research on attitude and value conflict ([14]; [39]). This approach was widely used to quantify the degree of conflict between value dimensions. Specifically, the formula for assessing conflicts between self-enhancement and self-transcendence is as follows: Value Conflict = (self-enhancement + self-transcendence)/2 − |self-transcendence − self-enhancement|. This formula captures both the degree of overlap and the divergence between these competing value dimensions.

SCC was measured using the 12-item Self-Concept Clarity Scale ([5]; [28]). This scale evaluates the clarity, consistency, and stability of an individual’s self-beliefs. Participants responded to items such as “I have a clear sense of who I am and what I stand for” and reverse-coded items like “Sometimes I feel that I am not really the person I appear to be”. Responses were recorded on a 5-point Likert scale ranging from 1 (“strongly disagree”) to 5 (“strongly agree”), with higher scores indicating greater SCC. Except for items 6 and 11, all other items on the scale were reverse-coded. These items were appropriately recoded before calculating the total score. The internal consistency of the scale in this study was acceptable, with a Cronbach’s α of 0.77.

Stress was assessed using the stress subscale of the Depression–Anxiety-Stress Scale (DASS-21) ([16]; [24]). This 7-item subscale measures the frequency and severity of stress-related symptoms over the past week. Participants rated each item on a 4-point Likert scale, ranging from 0 (“did not apply to me at all”) to 3 (“applied to me very much or most of the time”), with higher scores reflecting greater levels of perceived stress. There were no reverse-scored items on this subscale. Cronbach’s α for the stress subscale in this study was strong at 0.87, indicating high reliability.

Self-construal was measured using the Self-Construal Scale (SCS) ([22]; [36]), which distinguishes between independent and interdependent self-construal. The scale consists of 24 items, split evenly between 2 subscales. Participants rated their agreement with items on a 7-point Likert scale, ranging from 1 (“strongly disagree”) to 7 (“strongly agree”). The independent self-construal subscale measures autonomy and individuality (e.g., “I prefer to be self-reliant rather than depend on others”), while the interdependent self-construal subscale measures relational interdependence and social harmony (e.g., “My happiness depends on the happiness of those around me”). In this study, the internal consistency coefficients for the independent and interdependent subscales were α = 0.75 and α = 0.80, respectively. The Chinese version of SCS was widely validated and shown to have acceptable psychometric properties (Cronbach’s α = 0.88 for interdependence and α = 0.75 for independence).

### 2.5. Bias

To address common method bias, Harman’s single-factor test was performed. The analysis identified 13 distinct factors, with the largest factor accounting for 13.52% of the variance, well below the 40% threshold. Additionally, a single-factor CFA ([30]) was conducted, showing poor fit indices (χ^2^/df = 58.23; CFI = 0.21; TLI = 0.10; RMSEA = 0.27; SMRM = 0.13), indicating that common method bias was not a significant concern.

### 2.6. Statistical Methods

Data analysis was conducted using PROCESS version 4.0 (www.afhayes.com), accessed on 13 January 2025, and SPSS 21.0 ([17]). Preliminary analyses assessed the characteristics of the observed scales, including skewness, kurtosis, and adherence to normality assumptions ([11]; [38]). Relationships among variables were examined using Pearson correlation analysis. Mediating and moderated mediation effects were analyzed using PROCESS models 4 and 7 with 5000 bootstrap samples to estimate 95% bias-corrected confidence intervals. The independent variable was value conflict, the dependent variable was stress, SCC served as a mediating variable, and independent self-construal was the moderating variable.

## 3. Results

### 3.1. Descriptive Statistics and Correlation Analysis

Before conducting data analysis, systematic checks were performed to validate the model assumptions. First, the skewness and kurtosis coefficients for all variables were within acceptable ranges (skewness: −0.37 to 0.50; kurtosis: −0.47 to 0.17). Second, linear relationships were confirmed through scatterplots and residual plots, demonstrating linearity among all variables. Additionally, multicollinearity was assessed by calculating the Variance Inflation Factor (VIF) and tolerance values. All predictor variables had VIF values below 2 (range: 1.06–1.93) and tolerance values above 0.5, indicating that there were no significant multicollinearity issues.

As Table 1 shows, college students’ value conflicts between self-transcendence and self-enhancement were significantly and negatively correlated to SCC, and SCC was significantly and negatively correlated to stress. However, none of the above relationships were significant for the openness vs. conservation value conflicts. Therefore, in subsequent analyses, only the self-enhancement vs. self-transcendence value conflict was included as a dependent variable to test its effect on stress, along with the mediating role of SCC and the moderating role of self-construal concerning independence.

### 3.2. Testing the Mediating Role of Self-Concept Clarity

This study utilized Model 4 of PROCESS v4.0 to explore the mediating role of SCC in the relationship between value conflict and stress, while controlling for age and gender as covariates. The analysis revealed that value conflict significantly negatively predicted SCC (*β* = −0.08, *t* = −4.60, *p* < 0.001, 95% CI [−0.12, −0.05]), indicating that higher value conflict was associated with lower self-concept clarity. Additionally, SCC significantly negatively predicted stress (*β* = −0.65, *t* = −15.25, *p* < 0.001, 95% CI [−0.74, −0.57]), suggesting that greater SCC was linked to lower levels of stress. Furthermore, value conflict directly and positively predicted stress (*β* = 0.07, *t* = 3.26, *p* = 0.001, 95% CI [0.03, 0.11]), indicating that value conflict not only directly influenced stress but also had an indirect impact through its effect on SCC. The analysis of indirect effects further showed that the indirect effect of value conflict on stress via SCC was significant (indirect effect *β* = 0.05, Boot*SE* = 0.012, 95% BootCI [0.0290, 0.0761]), with a standardized indirect effect of 0.08 (Boot*SE* = 0.02, 95% BootCI [0.04, 0.12]). This suggests that SCC partially mediated the relationship between value conflict and stress, with the mediating effect accounting for 43.79% of the total effect. Covariate analysis indicated that age had a significant impact on stress (*β* = 0.03, *t* = 2.67, *p* = 0.008, 95% CI [0.01, 0.05]), while the effects of gender on SCC and stress were not significant. These findings suggest that the impact of value conflict on individual stress is partially mediated by the psychological mechanism of SCC. Improving SCC may therefore be an important intervention direction for alleviating stress.

### 3.3. Moderated Mediation Model Test

This study utilized Model 7 of PROCESS v4.0 to investigate the mediating role of SCC in the relationship between value conflict and stress, as well as the moderating effect of independent self-construal. The results (see Table 2) indicated that value conflict significantly negatively predicted SCC, and independent self-construal significantly moderated this relationship. The Johnson-Neyman test further revealed that under high levels of independent self-construal (+1 SD), the negative effect of value conflict on SCC was strongest (*β* = −0.13, *t* = −5.33, *p* < 0.001, 95% CI [−0.17, −0.08]), whereas under low levels of independent self-construal (−1 SD), this effect was not significant (*β* = −0.02, *t* = −0.83, *p* = 0.41) (see Figure 2).

Additionally, SCC significantly negatively predicted stress, and the direct effect of value conflict on stress was significant. The index of moderated mediation was significant (index = 0.0472, Boot*SE* = 0.0190, 95% BootCI [0.0093, 0.0842]), indicating that independent self-construal significantly moderated the indirect effect of value conflict on stress through SCC. When age and gender were included as covariates, the analysis showed that these covariates had minimal influence on the main effects. Specifically, after controlling for age and gender, the negative relationship between value conflict and SCC, as well as the moderating effect of independent self-construal, remained unchanged. Moreover, the negative predictive effect of SCC on stress was still significant. This suggests that while age and gender may have some impact on the model’s fit, they do not substantially alter the core relationships between variables. In summary, the findings support the hypothesized model, demonstrating that SCC is a crucial mediating mechanism between value conflict and stress (see Figure 3). Furthermore, independent self-construal moderates the impact of value conflict on SCC, thereby influencing the indirect effect of value conflict on stress through SCC. These findings provide valuable insights into the role of cultural dimensions on college students’ mental health and offer a theoretical basis for developing intervention strategies.

## 4. Discussion

This study investigated the interplay between value conflicts, stress, SCC, and self-construal among Chinese university students. The findings indicate that conflicts between self-transcendence and self-enhancement values are linked to reduced SCC, which subsequently associates with elevated stress levels. Notably, students exhibiting high levels of independent self-construal experience more pronounced negative effects, whereas those with lower levels are less vulnerable to the adverse impacts of value conflicts. These results highlight the critical role of self-construal in moderating the relationship between value conflicts and psychological well-being, suggesting that interventions aimed at enhancing self-concept clarity and fostering adaptive self-construal styles may be beneficial for students navigating conflicting values.

Swartz’s theory of value conflicts posits that inconsistent values can impede goal-directed behavior and generate approach-avoidance conflicts, potentially leading to psychopathological symptoms ([14]; [33]). Aligning with previous research, our findings demonstrate that conflicts between two values directly influence stress symptoms. This relationship may stem from the opposing motivational orientations of the conflicting values. Individuals striving to integrate both self-transcendence and self-enhancement values into their self-concept may frequently encounter internal conflicts, resulting in motivational tension with possible long-term detrimental effects, such as mental health issues ([2]; [6]). Research on Chinese college students reveals a concurrent pursuit of self-transcendence and self-enhancement values ([23]; [42]). However, limited studies have explored whether this simultaneous emphasis on conflicting values overwhelms young individuals, thereby increasing stress levels. The current study’s conclusions have significant implications for value education among university students, suggesting that emphasizing competing values might be detrimental to mental health. Therefore, educational programs should consider strategies to help students balance and integrate conflicting values to promote psychological well-being.

Interestingly, college students in this study did not report increased psychological stress from simultaneously pursuing openness and conservation values. This resilience may be attributed to the evolving context of Chinese society, where values have shifted significantly over the past four decades from traditional (e.g., respect for elders, obedience to authority, contentment) to modern ideals (e.g., affirmativeness, open-mindedness, independence, optimism) ([4]). According to Hofstede’s cultural dimensions, China scores high on collectivism, which emphasizes group harmony and interdependence, facilitating the coexistence of conservation and openness values ([20]). Consequently, conservation and openness values may coexist harmoniously within the Chinese cultural framework ([44]). Additionally, modern Chinese students often view openness as the normative choice, as educational institutions and parents encourage freedom, self-expression, and self-discovery ([41]). Supporting this, prior research indicates that Chinese university students prioritize openness over conservation ([23]). Moreover, studies suggest that values with opposing motivations can be synergistic when combined to achieve overarching goals ([41]). This compatibility implies that in certain cultural contexts, the simultaneous pursuit of seemingly conflicting values does not necessarily lead to increased stress, thereby informing the development of culturally sensitive mental health interventions that acknowledge and incorporate these value dynamics.

The findings reveal an indirect relationship between increased stress and high levels of value conflict, with reduced SCC serving as a mediating factor. These results align with those of [14] ([14]), enhancing our understanding of the mediating mechanisms linking stress and value conflicts among Chinese university students. Individuals’ self-identification and self-perception are significantly influenced by their values ([1]; [40]). When a value is integrated into an individual’s self-concept, it becomes a core component of their identity ([45]). Therefore, conflicts between values often involve self-involvement processes, where trade-offs necessitate adopting a self-identity that may undermine the individual’s pursuit of self-consistency ([19]). This can lead to conflicting self-concepts, thereby intensifying negative emotions such as stress ([14]). In summary, this study offers deeper insights into the psychological mechanisms by which value conflicts affect stress perceptions, reinforcing the robust connection between values and self-concept. These insights can inform the creation of future interventions aimed at addressing psychological issues arising from value conflicts, such as developing programs that enhance self-concept clarity and promote adaptive self-construal styles to mitigate stress.

Lastly, the primary aim of this paper was to investigate whether independent self-construal moderates the indirect relationship between SCC and value conflict. The study confirmed the presence of this moderating effect. Given the rising prominence of independent self-construal among contemporary Chinese college students, it is pertinent to explore how this self-construal dimension influences the relationship between SCC and value conflict. Individuals with higher independent self-construal scores tend to be more individualistic, possess less dialectical thinking, and maintain a more stable self-concept ([9]; [10]; [13]). Consequently, the perceived consistency and integrity of their self-concept are more easily disrupted when confronted with value conflicts, resulting in heightened psychological pressure when self-concept clarity diminishes. In contrast, East Asian cultures, as defined by Hofstede’s cultural dimensions theory, emphasize collectivism and interdependence ([20]). Collectivism fosters a sense of community and interconnectedness, making individuals from these cultures more adaptable and tolerant of value conflicts due to a more flexible and less rigid self-concept ([8]; [31]). Supporting this, [42] ([42]) found that some Chinese college students scored high on both opposing value dimensions without experiencing significant value conflicts. Xie et al. attributed this to the influence of Confucianism’s philosophy of mediocrity, which fosters dialectical thinking and enables the integration of individual and social orientations, as well as traditional and modern ideals, without conflict. This largely explains why individuals with lower levels of independent self-construal do not experience significant declines in self-concept clarity when faced with value conflicts ([37]). Future research should incorporate cross-cultural studies to further validate these conclusions and explore the role of different self-construal dimensions in various cultural contexts.

Despite the significant findings, this study has several limitations. Firstly, the cross-sectional design limits the ability to infer causal relationships between value conflict, SCC, and stress. Future research should employ longitudinal designs to establish temporal precedence and clarify causal links between these variables. Secondly, the reliance on self-reported measures may introduce biases such as social desirability or inaccurate self-perception. Future studies could reduce these biases by incorporating multiple data sources, such as behavioral assessments or peer/mentor reports, which would enhance the validity of the findings. The generalizability of the results is another limitation, as our sample primarily consisted of university students from a single public university in China. Future research should include more diverse and larger samples from various cultural backgrounds and demographics to examine the applicability of the findings across different populations. Finally, this study did not conduct sensitivity analyses to assess the robustness of the results. Future studies could explore alternative model specifications, such as including additional covariates or testing different moderators, to further validate the robustness of the proposed framework.

The findings of this study have several practical implications for mental health interventions targeting university students. Universities should develop counseling programs that specifically address value conflicts, helping students navigate the tensions between self-transcendence and self-enhancement values. By incorporating strategies that enhance self-concept clarity and promote adaptive self-construal styles, counselors can assist students in reducing stress associated with conflicting values. Additionally, integrating value education into university curriculum can help students develop a balanced understanding of competing values. Educational initiatives should emphasize the coexistence of self-transcendence and self-enhancement values, teaching students how to harmonize these values to maintain psychological well-being. Hosting workshops and seminars that focus on self-construal and self-concept clarity can equip students with the skills to manage value conflicts effectively. These sessions can include activities that promote dialectical thinking and resilience, fostering a more flexible self-concept that can withstand value-induced stress. Establishing peer support groups where students can share their experiences with value conflicts can also create a supportive environment, providing mutual support and practical solutions for managing stress related to conflicting values. Finally, collaboration between universities and mental health professionals is essential to design culturally sensitive interventions that resonate with students’ cultural backgrounds and personal experiences, ensuring the effectiveness and acceptance of these programs.

## 5. Conclusions and Implications

Using Chinese college students as subjects, this study investigated the relationships linking two types of value conflicts to stress perception within the framework of Schwartz’s value theory. We also looked more closely at the moderating function of self-construal of independence and the mediating role of self-concept clarity. Our findings show that stress perceptions in college students were positively influenced by conflict between two values, but not by conflict between openness and conservatism. This impact might result from making people’s self-concept less clear. However, this psychological mechanism was stronger in subjects with high self-construal of independence, while for the subjects with low self-construal of independence, value conflict did not significantly weaken their level of self-concept clarity.

The study’s conclusions hold significant implications for both clinical practice and academic research. Firstly, there is a growing emphasis in clinical and psychological therapy disciplines on the relationship between value conflicts and mental health ([2]; [42]). This study demonstrates that conflicts between self-transcendence and self-enhancement values impose greater strain on students compared to conflicts between openness and conservation values. These insights can inspire the development of targeted value education programs and counseling strategies for university students, helping mental health professionals design interventions that specifically address and mitigate the stress arising from conflicting values. Secondly, this research advances earlier studies by delving deeper into the psychological mechanisms through which value conflicts lead to increased stress perception, highlighting the roles of self-construal and SCC. From the perspective of self-concept, this framework can guide future interventions aimed at reducing the negative impacts of value conflicts by enhancing SCC and promoting adaptive self-construal styles among students. Additionally, the study suggests that cultural context plays a crucial role in how individuals manage value conflicts, as self-concept stability and consistency are valued differently across cultures. Specifically, individuals from collectivist cultures may experience value conflicts less intensely than those from individualistic cultures. This underscores the necessity for culturally tailored approaches in value education and mental health interventions. Future research should therefore conduct more cross-cultural studies to examine these differences and validate the applicability of the current findings in diverse cultural settings.

## Figures and Tables

**Figure 1 behavsci-15-00104-f001:**
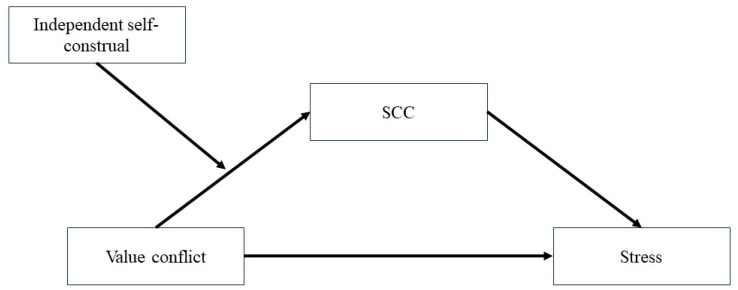
Proposed moderated mediation model.

**Figure 2 behavsci-15-00104-f002:**
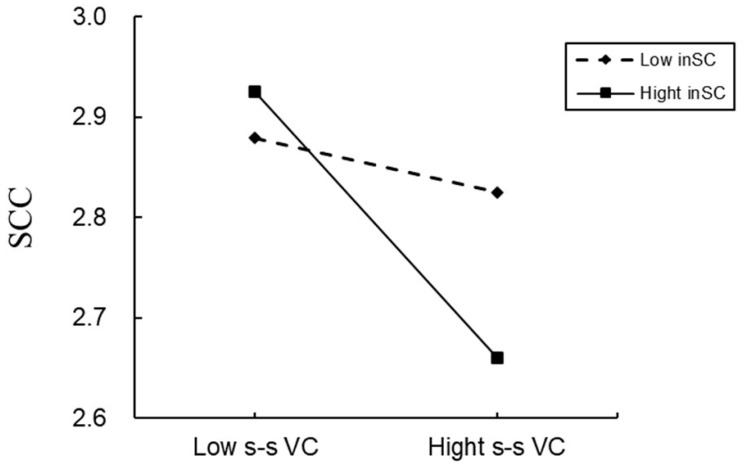
Interaction between self-transcendence vs. self-enhancement value conflicts and independent self-construal on SCC.

**Figure 3 behavsci-15-00104-f003:**
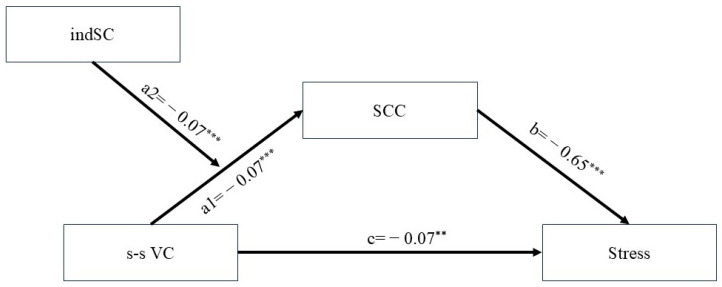
Moderated mediation model and path coefficients. ** *p* < 0.01, *** *p* < 0.001.

**Table 1 behavsci-15-00104-t001:** Descriptive statistics and correlations for all variables.

	M	SD	s-s VC	o-c VC	Stress	inSC	inerSC	SCC
s-s VC	3.57	1.01	—					
o-c VC	3.61	0.90	0.37 ***	—				
stress	1.94	0.70	0.19 ***	−0.03	—			
indSC	4.91	0.73	0.16 ***	0.30 ***	−0.05	—		
inerSC	5.04	0.78	0.09 *	0.08 *	0.03	0.14 ***	—	
SCC	2.83	0.52	−0.16 **	−0.02	−0.51 ***	−0.02	0.00	—

s-s VC = self-transcendence vs. self-enhancement value conflict; o-c VC = openness vs. conservation value conflict; indSC = independent self-construal; interSC = interdependent self-construal; SCC = self-concept clarity. * *p* < 0.05, ** *p* < 0.01, *** *p* < 0.001.

**Table 2 behavsci-15-00104-t002:** Testing the moderated mediation effect.

		Model 1			Model 2	
		SCC			Stress	
	*β*	*t*	95% CI	*β*	*t*	95% CI
gender	−0.06	−1.47	[−0.15, 0.02]	0.01	0.26	[−0.08, 0.11]
age	0.00	0.31	[−0.14, 0.02]	0.03	2.67 **	[0.01, 0.05]
s-s VC	−0.07	−4.15 ***	[−0.11, −0.04]	0.07	3.26 *	[0.03, 0.11]
indSC	−0.00	−0.04	[−0.05, 0.05]			
s-s VC × inSC	−0.07	−2.93 ***	[−0.12, −0.02]			
SCC				−0.65	−15.25 ***	[−0.74, −0.57]
*R* ^2^	0.04				0.27	
*F*	6.70 ***				69.48 ***	

s-s VC = self-transcendence vs. self-enhancement value conflict; indSC = independent self-construal; SCC = self-concept clarity. Gender is encoded as dummy variable with 1 = male, 0 = female. All *β*s in this table are unstandardized coefficients. * *p* < 0.05, ** *p* < 0.01, *** *p* < 0.001.

## Data Availability

No new data were generated or analyzed in this study. The data supporting the findings of this research are not publicly available due to privacy or ethical restrictions, but are available upon reasonable request from the corresponding author.

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
