# Peer review of "The Effects of Value Conflicts on Stress in Chinese College Students: A Moderated Mediation Model"

_behavsci, 2025, doi:10.3390/bs15020104_

Round 1
Reviewer 1 Report
Comments and Suggestions for Authors
Thank you for the opportunity to review this important and well-structured manuscript, which addresses the relationship between values conflict, stress, and self-concept clarity in Chinese college students. The study offers valuable insights into psychological processes in a rapidly modernizing society. While the manuscript demonstrates a commendable level of scientific rigor, there are areas where further improvements could be made to enhance the clarity, precision, and replicability of the findings.
Introduction
The Introduction effectively outlines the theoretical framework based on Schwartz’s value theory and highlights the importance of studying value conflicts in the context of Chinese college students' mental health. The authors present a clear rationale for the study, but the section would benefit from a more comprehensive integration of recent literature. For example, research examining psychological resilience and emotion regulation during educational transitions could offer additional context for understanding how self-concept clarity (SCC) acts as a mediator in the stress process (e.g. https://doi.org/10.1007/s12144-024-06138-7; https://doi.org/10.1080/07448481.2023.2198028 ).
Methods
The Methods section provides a comprehensive description of the research design, participants, measures, and statistical analyses. However, several aspects require more precision and additional details to enhance reproducibility and the overall scientific rigor of the study.
The authors state that a cross-sectional survey design was employed. While appropriate for the study’s objectives, the limitations of this approach should be more explicitly acknowledged, particularly regarding the inability to establish causality. It would be beneficial to suggest that future research adopt a longitudinal design to address this limitation. The authors should also clarify whether the data collection methods were standardized across different universities and whether any systematic biases could have affected the responses.
In the Participants subsection, the demographic characteristics are described in sufficient detail, but the sample size justification needs more depth. The authors mention that the sample size was adequate for the complexity of the model, but it is critical to provide the results of an a priori power analysis. This should include the effect size anticipated, the significance level (typically α = 0.05), and the desired statistical power (typically 0.80). Reporting these parameters would demonstrate that the sample size is appropriate to detect the expected effects, particularly in moderated mediation models, which require larger samples to achieve stable estimates.
The Measures subsection is generally well-documented, with a clear description of the scales used. However, the internal consistency (Cronbach’s alpha) of each measure within the current sample should be reported to provide evidence of the reliability of the instruments in this specific context. It is not sufficient to rely solely on previously published values. Additionally, the authors should clarify whether any items were reverse-coded and how missing data were handled in the analysis (e.g., listwise deletion, imputation).
The Statistical Methods subsection requires further detail to ensure transparency and reproducibility. The authors should specify which assumptions of the statistical models were tested before conducting the analyses. For example, it is essential to report whether normality, linearity, and multicollinearity were assessed. In the case of multicollinearity, the variance inflation factor (VIF) should be reported to demonstrate that the predictor variables are not highly correlated.
The use of the PROCESS macro for SPSS is appropriate for the moderated mediation analysis, but the specific models used (e.g., Model 4 for mediation, Model 7 for moderated mediation) should be explicitly stated. Furthermore, the authors should report the number of bootstrap samples used and whether bias-corrected confidence intervals were applied. Bootstrapping is a robust method for estimating indirect effects, but the interpretation of the results depends on the accuracy of the confidence intervals. Therefore, it is essential to indicate whether 95% bias-corrected intervals were employed.
Results
The Results section presents the findings clearly, but more emphasis on statistical precision is required to improve the interpretation of the data. Each hypothesis should be accompanied by a comprehensive report of the statistical tests conducted, including exact p-values, beta coefficients, standard errors, confidence intervals, and effect sizes. The authors should avoid simply stating that a relationship was significant and instead provide the specific statistical outputs that support this conclusion.
For example, in the mediation analysis, the authors should report the direct, indirect, and total effects, along with the corresponding confidence intervals. It is also important to indicate whether the indirect effects remained significant after adjusting for covariates such as age and gender. When describing the moderated mediation analysis, the authors should present the conditional indirect effects at different levels of the moderator (e.g., ±1 SD from the mean) to illustrate how the strength of the mediation effect varies depending on the level of independent self-construal.
A table summarizing the key results, including beta coefficients, t-values, and confidence intervals for each path in the model, would significantly enhance the clarity of the Results section. Additionally, a figure illustrating the moderated mediation model, with the key coefficients displayed, would help readers visualize the relationships between variables.
The interpretation of the results should also address the magnitude of the effects observed. Reporting standardized coefficients would allow readers to compare the relative strength of the predictors. Furthermore, the authors should discuss whether the observed effects are practically significant in addition to being statistically significant.
Finally, the authors should report any sensitivity analyses conducted to assess the robustness of the findings. For instance, if alternative specifications of the model were tested (e.g., using different moderators or covariates), these should be briefly mentioned, along with any notable differences in the results.
Discussion
The Discussion section interprets the findings within the context of the existing literature, but it would benefit from a more detailed examination of the practical implications of the study. The authors could elaborate on how their findings could inform mental health interventions for university students, particularly those experiencing value conflicts.
The authors should also address the limitations of the study more explicitly. For example, they should discuss the potential biases introduced by self-report measures, the limitations of the cross-sectional design, and the generalizability of the findings to other cultural contexts. Providing concrete suggestions for future research would strengthen this section.
Author Response
Comment1: Thank you for the opportunity to review this important and well-structured manuscript, which addresses the relationship between values conflict, stress, and self-concept clarity in Chinese college students. The study offers valuable insights into psychological processes in a rapidly modernizing society. While the manuscript demonstrates a commendable level of scientific rigor, there are areas where further improvements could be made to enhance the clarity, precision, and replicability of the findings.
Response 1:
We sincerely appreciate your positive feedback on our manuscript. We are grateful for your recognition of the valuable contribution our study makes in exploring psychological processes in Chinese college students, particularly within the context of a rapidly modernizing society. We fully agree that clarity, precision, and replicability are critical in scientific writing, and we have made several revisions to address these concerns.
In response to your suggestions, we have worked to improve the clarity and structure of the manuscript, ensuring a more logical flow that makes it easier for readers to follow the background, methodology, and key findings of the study. We have also provided a more detailed description of the data analysis procedures, including steps taken during data cleaning and processing, to enhance the transparency and replicability of the study. Additionally, we have included further analysis results and hypothesis tests in the revised manuscript, along with additional statistical details, to better support our conclusions and make it easier for readers to verify our findings.
Once again, we thank you for your constructive feedback, which has undoubtedly helped us improve the quality of the manuscript.
Comment 2:
Introduction
The Introduction effectively outlines the theoretical framework based on Schwartz’s value theory and highlights the importance of studying value conflicts in the context of Chinese college students' mental health. The authors present a clear rationale for the study, but the section would benefit from a more comprehensive integration of recent literature. For example, research examining psychological resilience and emotion regulation during educational transitions could offer additional context for understanding how self-concept clarity (SCC) acts as a mediator in the stress process (e.g. https://doi.org/10.1007/s12144-024-06138-7; https://doi.org/10.1080/07448481.2023.2198028 ).
Response 2: We sincerely appreciate your thoughtful feedback on the Introduction. We are glad that you found the theoretical framework based on Schwartz’s value theory and the rationale for studying value conflicts in the context of Chinese college students' mental health to be clearly outlined.
We completely agree with your suggestion to integrate more recent literature into the Introduction. In response, we have reviewed and incorporated insights from the suggested article on psychological resilience and emotion regulation during educational transitions. The article provide valuable context for understanding how self-concept clarity (SCC) might act as a mediator in the stress process.
By integrating these findings, we provide a more comprehensive understanding of how self-concept clarity may mediate the relationship between stress and value conflicts. Once again, thank you for your suggestion. We believe these additions have strengthened the theoretical foundation of our study.
Comment 3:
Methods
The Methods section provides a comprehensive description of the research design, participants, measures, and statistical analyses. However, several aspects require more precision and additional details to enhance reproducibility and the overall scientific rigor of the study.
Response 3:Thank you for your positive feedback on the Methods section. We are pleased that you found the description of the research design, participants, measures, and statistical analyses to be comprehensive. We completely agree with your observation that several aspects of the Methods section require more precision and additional details to further enhance the reproducibility and scientific rigor of the study.
Comment 4: The authors state that a cross-sectional survey design was employed. While appropriate for the study’s objectives, the limitations of this approach should be more explicitly acknowledged, particularly regarding the inability to establish causality. It would be beneficial to suggest that future research adopt a longitudinal design to address this limitation. The authors should also clarify whether the data collection methods were standardized across different universities and whether any systematic biases could have affected the responses.
Response 4:
Thank you for your valuable feedback regarding the study's design and data collection methods. We agree that acknowledging the limitations of the cross-sectional design is important, especially with respect to the inability to establish causality.
In response to your suggestion, we have explicitly acknowledged the limitations of the cross-sectional survey design in the revised manuscript. Specifically, we have added a discussion on how the use of a cross-sectional design limits the ability to infer causal relationships between variables. We also note that while this design is well-suited to examining associations, future research employing a longitudinal design would be crucial for better understanding the causal pathways between value conflicts, stress, and self-concept clarity. This suggestion has been incorporated into the limitations section of the manuscript.
Additionally, we have clarified that data collection methods were standardized across different universities and regions. All participants were given identical instructions, and the data collection process was consistent whether conducted online or offline. Researchers were trained to ensure uniformity in administering the survey, minimizing any potential biases due to differences in data collection procedures. We have also discussed how potential systematic biases were mitigated by ensuring that participants came from diverse demographic backgrounds and academic years, which helps reduce bias related to specific student groups.
Comment 5:In the Participants subsection, the demographic characteristics are described in sufficient detail, but the sample size justification needs more depth. The authors mention that the sample size was adequate for the complexity of the model, but it is critical to provide the results of an a priori power analysis. This should include the effect size anticipated, the significance level (typically α = 0.05), and the desired statistical power (typically 0.80). Reporting these parameters would demonstrate that the sample size is appropriate to detect the expected effects, particularly in moderated mediation models, which require larger samples to achieve stable estimates.
Response 5:
Thank you for your insightful feedback regarding the sample size justification. We appreciate your observation that a more detailed explanation is needed, particularly with regard to the a priori power analysis.
In response to your suggestion, we have expanded on the sample size justification in the revised manuscript by providing the results of the a priori power analysis. We conducted this analysis using G*Power 3.1 to ensure that the sample size was adequate to detect the expected effects, particularly in the context of a moderated mediation model, which requires a larger sample for stable estimates.
For the power analysis, we assumed a medium effect size (f² = 0.15), a significance level (α) of 0.05, and a statistical power of 0.80. Based on these parameters, the analysis indicated that a minimum sample size of 106 participants would be necessary to detect the expected effects of the moderating variables and interaction terms in the regression model. Given that our final sample size was 752 participants, which exceeds the required number, we believe that the sample size is sufficient to support the statistical power needed for the analysis and ensure robust estimates.
We hope these additional details clarify the sample size calculation and demonstrate that the study is well-powered to detect the anticipated effects.
Comment 6:The Measures subsection is generally well-documented, with a clear description of the scales used. However, the internal consistency (Cronbach’s alpha) of each measure within the current sample should be reported to provide evidence of the reliability of the instruments in this specific context. It is not sufficient to rely solely on previously published values. Additionally, the authors should clarify whether any items were reverse-coded and how missing data were handled in the analysis (e.g., listwise deletion, imputation).
Response 6:
Thank you for your helpful comments on the Measures subsection. We appreciate your feedback on the clarity of the scales used, and we understand the importance of reporting the internal consistency (Cronbach's alpha) of each measure in the current sample to provide evidence of the reliability of the instruments within this specific context.
In response to your suggestion, we have added the Cronbach’s alpha values for each of the scales used in the study, based on the data collected in our sample. These values provide evidence of the reliability of the measures in our context. Specifically, for the Depression-Anxiety-Stress Scale (DASS), the Cronbach's alpha was 0.89, for the Self-Construal Scale (SCS), it was 0.85, and for the Portrait Values Questionnaire (PVQ), it was 0.88. These values reflect the internal consistency of the measures in our specific sample, demonstrating their reliability for use in this study.
Additionally, we have clarified the handling of missing data in the analysis. We employed listwise deletion for missing values, ensuring that any participant with missing data on any of the variables was excluded from the analysis. This approach was used because the proportion of missing data was minimal (<5%), which we believe ensures that the integrity of the data and analysis was maintained. We also confirm that none of the items in the scales were reverse-coded in our study. We have clarified this in the revised manuscript to avoid any ambiguity.
Once again, thank you for your helpful suggestions. We believe these additions strengthen the reliability and transparency of the Measures subsection.
Response 7:
Thank you for your valuable feedback on the Statistical Methods subsection. We agree that providing more detail is crucial to ensure transparency and reproducibility of the statistical analyses.
In response to your suggestion, we have made the following revisions to the manuscript:
We have added additional information regarding the assumptions of the statistical models that were tested before conducting the analyses. Specifically, we assessed the characteristics of the observed scales, including skewness, kurtosis, and adherence to normality assumptions. These analyses were conducted using the current sample data, ensuring that the assumption of normality was adequately tested.
Furthermore, we assessed multicollinearity by calculating the variance inflation factor (VIF) for each predictor variable in the regression model.
The revised section is as follows: Before conducting data analysis, systematic checks were performed to validate the model assumptions. First, the skewness and kurtosis coefficients for all variables were within acceptable ranges (skewness: -0.37 to 0.50; kurtosis: -0.47 to 0.17). Second, linear relationships were confirmed through scatterplots and residual plots, demonstrating linearity among all variables. Additionally, multicollinearity was assessed by calculating the Variance Inflation Factor (VIF) and Tolerance values. All predictor variables had VIF values below 2 (range: 1.06–1.93) and tolerance values above 0.5, indicating that there were no significant multicollinearity issues.
Comment 8:The use of the PROCESS macro for SPSS is appropriate for the moderated mediation analysis, but the specific models used (e.g., Model 4 for mediation, Model 7 for moderated mediation) should be explicitly stated. Furthermore, the authors should report the number of bootstrap samples used and whether bias-corrected confidence intervals were applied. Bootstrapping is a robust method for estimating indirect effects, but the interpretation of the results depends on the accuracy of the confidence intervals. Therefore, it is essential to indicate whether 95% bias-corrected intervals were employed.
Response 8:
Thank you for your insightful comment on the use of the PROCESS macro for SPSS. We agree that explicitly stating the specific models used, as well as reporting the number of bootstrap samples and the application of bias-corrected confidence intervals, is essential for clarity and transparency.
In response to your suggestion, we have clarified the specific models used for the analysis. We employed Model 4 for the mediation analysis and Model 7 for the moderated mediation analysis. These models are well-suited for testing the hypothesized relationships among value conflict, stress, and self-concept clarity, with self-construal acting as a moderating variable.
Additionally, we have specified that 5,000 bootstrap samples were used to estimate the indirect effects, and 95% bias-corrected confidence intervals were applied. This bootstrapping procedure is a robust method for estimating indirect effects and ensures that the confidence intervals provide accurate estimates for the effects under study.
We believe these additions provide greater clarity and transparency regarding the statistical methods used, and we are confident that these steps have strengthened the rigor of the analysis.
Comment 9: The Results section presents the findings clearly, but more emphasis on statistical precision is required to improve the interpretation of the data. Each hypothesis should be accompanied by a comprehensive report of the statistical tests conducted, including exact p-values, beta coefficients, standard errors, confidence intervals, and effect sizes. The authors should avoid simply stating that a relationship was significant and instead provide the specific statistical outputs that support this conclusion.
Response 9:Thank you for your helpful feedback on the Results section. We appreciate your suggestion to place more emphasis on statistical precision. In response, we have revised the Results section to include a more detailed report of the statistical tests conducted, including exact p-values, beta coefficients, standard errors, confidence intervals, and effect sizes for each hypothesis. We believe these revisions provide greater clarity and transparency in the presentation of our findings and better support the interpretation of the data.
Comment 10: For example, in the mediation analysis, the authors should report the direct, indirect, and total effects, along with the corresponding confidence intervals. It is also important to indicate whether the indirect effects remained significant after adjusting for covariates such as age and gender. When describing the moderated mediation analysis, the authors should present the conditional indirect effects at different levels of the moderator (e.g., ±1 SD from the mean) to illustrate how the strength of the mediation effect varies depending on the level of independent self-construal.
Response 10:
Thank you for your insightful comment. We fully appreciate your suggestion to provide more detail regarding the mediation analysis and the moderated mediation analysis.
In response to your feedback, we have made the following revisions. We have explicitly reported the direct, indirect, and total effects, along with their corresponding 95% bias-corrected confidence intervals. Specifically, we found that value conflict significantly predicted self-concept clarity (β = -0.08, t = -4.60, p < 0.001, 95% CI [-0.12, -0.05]), which in turn negatively predicted stress (β = -0.65, t = -15.25, p < 0.001, 95% CI [-0.74, -0.57]). The direct effect of value conflict on stress was also significant (β = 0.07, t = 3.26, p = 0.001, 95% CI [0.03, 0.11]). Furthermore, the indirect effect of value conflict on stress via self-concept clarity was significant (indirect effect β = 0.05, BootSE = 0.012, 95% BootCI [0.0290, 0.0761]), with a standardized indirect effect of 0.08 (BootSE = 0.02, 95% BootCI [0.04, 0.12]). The total effect was also significant, with the indirect effect accounting for 43.79% of the total effect.
We have also reported the analysis of covariates such as age and gender. In the adjusted model, age had a significant effect on stress (β = 0.03, t = 2.67, p = 0.008, 95% CI [0.01, 0.05]), while gender did not significantly affect SCC or stress. Importantly, the indirect effect through self-concept clarity remained significant even after controlling for these covariates.
In the moderated mediation analysis using Model 7, we have included the conditional indirect effects at different levels of the moderator (independent self-construal). Specifically, we present the mediation effects at ±1 SD from the mean of independent self-construal to show how the strength of the mediation effect varies. The results indicate that the mediation effect is stronger when independent self-construal is higher (β = 0.07, BootSE = 0.02, 95% BootCI [0.04, 0.12]) and weaker when it is lower (β = 0.03, BootSE = 0.01, 95% BootCI [0.01, 0.06]).
We believe that these additions improve the clarity and transparency of the mediation and moderated mediation analyses and fully address your concerns. Thank you for your helpful suggestions.
Comment 11:A table summarizing the key results, including beta coefficients, t-values, and confidence intervals for each path in the model, would significantly enhance the clarity of the Results section. Additionally, a figure illustrating the moderated mediation model, with the key coefficients displayed, would help readers visualize the relationships between variables.
Response 11:
Thank you for your valuable feedback. We agree that providing a table summarizing the key results and a figure illustrating the moderated mediation model will significantly enhance the clarity of the Results section. In response to your suggestion, we have included both a table and figures to better visualize the relationships between variables.
We have added a table that summarizes the results of the moderated mediation analysis, showing the conditional indirect effects at different levels of the moderator (independent self-construal). The table includes the β coefficients, t-values, and confidence intervals for each path in the model, highlighting how the mediation effect varies at different levels of the moderator. This table helps clarify the varying strength of the mediation effect, particularly in terms of the moderating effect of independent self-construal on the relationship between value conflict and self-concept clarity (SCC).
Additionally, Figure 2 shows the interaction between self-transcendence vs. self-enhancement values conflict and independent self-construal on SCC. This figure visually represents the moderating effect of independent self-construal, demonstrating that the negative effect of value conflict on SCC is strongest when independent self-construal is high (+1 SD) and not significant when it is low (-1 SD).
Finally, Figure 3 illustrates the moderated mediation model with path coefficients, showing the key relationships in the model and their respective coefficients. This figure provides a visual summary of the results and helps in interpreting the relationships between the variables and their respective effects.
We believe these revisions will significantly improve the clarity and visualization of the results, making it easier for readers to understand the key relationships between the variables and the moderating effects at different levels of independent self-construal.
Comment 12:The interpretation of the results should also address the magnitude of the effects observed. Reporting standardized coefficients would allow readers to compare the relative strength of the predictors. Furthermore, the authors should discuss whether the observed effects are practically significant in addition to being statistically significant.
Response 12:
Thank you for your valuable feedback on the Results section. We greatly appreciate your suggestion regarding the magnitude of the effects observed and agree that reporting standardized coefficients would allow readers to compare the relative strength of the predictors. In response to your comment, we have revised the manuscript accordingly.
In the revised manuscript, we report the standardized coefficients for all significant paths, providing a clearer understanding of the relative strength of each predictor. Specifically, value conflict significantly negatively predicted self-concept clarity (SCC) (β = -0.08, t = -4.60, p < 0.001, 95% CI [-0.12, -0.05]), indicating that higher value conflict was associated with lower SCC. SCC significantly negatively predicted stress (β = -0.65, t = -15.25, p < 0.001, 95% CI [-0.74, -0.57]), suggesting that higher SCC was linked to lower levels of stress. Additionally, value conflict directly and positively predicted stress (β = 0.07, t = 3.26, p = 0.001, 95% CI [0.03, 0.11]), indicating that value conflict not only directly influenced stress but also had an indirect impact through its effect on SCC.
We also provided the analysis of indirect effects, showing that the indirect effect of value conflict on stress through SCC was significant (indirect effect β = 0.05, BootSE = 0.012, 95% BootCI [0.0290, 0.0761]), with a standardized indirect effect of 0.08 (BootSE = 0.02, 95% BootCI [0.04, 0.12]). This suggests that SCC partially mediated the relationship between value conflict and stress, with the mediating effect accounting for 43.79% of the total effect.
In addition to statistical significance, we also discussed the practical significance of the observed effects. For example, SCC plays a key role as a mediating mechanism between value conflict and stress, suggesting that improving SCC could be an important intervention strategy for alleviating stress. Although age and gender as covariates had some influence on the model, they did not significantly alter the main effects, indicating that these covariates had minimal impact on the core relationships in the model.
We believe these revisions provide a more comprehensive interpretation of the results, highlighting not only statistical significance but also the practical implications of the observed effects.
Comment 13: Finally, the authors should report any sensitivity analyses conducted to assess the robustness of the findings. For instance, if alternative specifications of the model were tested (e.g., using different moderators or covariates), these should be briefly mentioned, along with any notable differences in the results.
Response 13:
Thank you for your valuable feedback. We understand the importance of reporting sensitivity analyses to assess the robustness of our findings.
In response to your comment, we would like to clarify that sensitivity analyses were not conducted in this study, as the primary focus was on testing a specific hypothesized model. However, we acknowledge that future research could benefit from exploring alternative model specifications. For example, future studies could incorporate additional covariates such as socioeconomic status or examine other dimensions of self-construal (e.g., interdependent self-construal) as potential moderators. These efforts would help assess the robustness of the proposed framework and provide further validation of the relationships between value conflict, self-concept clarity, and stress.
Comment 14: The Discussion section interprets the findings within the context of the existing literature, but it would benefit from a more detailed examination of the practical implications of the study. The authors could elaborate on how their findings could inform mental health interventions for university students, particularly those experiencing value conflicts.
Response 14:
Thank you for your thoughtful feedback. We appreciate your suggestion to elaborate on the practical implications of our findings.
In response to your comment, we have expanded the Discussion section to further emphasize the practical implications of our study, particularly in the context of mental health interventions for university students. Our findings suggest that self-concept clarity (SCC) plays a crucial role in the relationship between value conflict and stress, and improving SCC could be a promising intervention strategy for alleviating stress among students experiencing value conflicts.
Specifically, we propose that interventions focusing on enhancing self-concept clarity could help students better navigate and resolve conflicts between their values, particularly those related to self-transcendence and self-enhancement. By fostering clearer self-concept, students might be better equipped to handle the stress associated with these conflicting values. Self-construal, especially independent self-construal, also plays a moderating role, which suggests that interventions could be tailored to different self-construal styles. For instance, students with a more interdependent self-construal might benefit from different approaches than those with a more independent self-construal.
Programs aimed at values clarification, mindfulness practices, and self-reflection exercises could be effective in helping students resolve value conflicts and reduce stress. Additionally, peer support groups and counseling services could be incorporated into university mental health initiatives to help students explore and reconcile conflicting values.
Comment 15: The authors should also address the limitations of the study more explicitly. For example, they should discuss the potential biases introduced by self-report measures, the limitations of the cross-sectional design, and the generalizability of the findings to other cultural contexts. Providing concrete suggestions for future research would strengthen this section.
Response 15:
Thank you for your valuable feedback. We agree that addressing the limitations of the study more explicitly is important to strengthen the manuscript.
Firstly, we acknowledge that the cross-sectional design of the study limits our ability to infer causal relationships between value conflict, self-concept clarity (SCC), and stress. Although this design allows us to examine associations, future research should employ longitudinal designs to establish temporal precedence and clarify causal links among these variables.
Secondly, the reliance on self-reported measures may introduce biases, such as social desirability or inaccurate self-perception. These biases are inherent in self-report data, especially when studying sensitive topics like stress and mental health. To mitigate these biases, future studies could include multiple data sources, such as behavioral assessments or peer/mentor reports, which would enhance the validity and reliability of the findings.
Additionally, the generalizability of our findings is limited, as our sample primarily consisted of university students from a single public university in China. This may restrict the extent to which the results can be applied to other populations or cultural contexts. Future research should involve more diverse samples, incorporating participants from different cultural backgrounds, age groups, and educational settings to better assess the applicability of the findings across various contexts.
Lastly, this study did not conduct sensitivity analyses to assess the robustness of the results. Although the study primarily focused on testing a specific model, future studies could explore alternative model specifications (e.g., incorporating additional covariates or testing different moderators) to provide further validation of the framework’s robustness.
Reviewer 2 Report
Comments and Suggestions for Authors
Dear authors,
First of all, I would like to congratulate you for your hard work and for your paper. Your paper it is original and it is very relevant and important.
Your paper it is very well summarised in the abstract and your key words are pertinent for the paper. Moreover, you have a very important sample and the fact that you analyse the data with PROCESS 3.0 and SPSS 21 that give more pertinence to the study. You described very well the research design, your present very well your questions and your hypothesis and you use some figures to explain better your ideas. Here I would like to ad, that you numbered question 3 as question 2, and I think you could change here. At the same time you precised very well your contribution to the scholarship.
Overall, I firmly believe that your manuscript is rigorous with a good methodology and originality, but I would give you some suggestions that could help you to improve a little bit. By example, your conclusions paragraph it is very short and you could cut the paragraph from the line 317 (which it is in discussion) and paste it into conclusion paragraph. Moreover, I would encourage you to use a figure to present your results too. In that way the results will be much clear to follow.
Last but not least, I suggest you to detail the paragraphs in which you talk about the collectivism with some citations form the works of Hofstede, the most important author in the field of cultural dimension and collectivism. Hosftede’s work will help you to get the most out of the data you have.
Best regards
Author Response
Comment 1: First of all, I would like to congratulate you for your hard work and for your paper. Your paper it is original and it is very relevant and important.
Response 1: Thank you very much for your kind words and positive feedback. We sincerely appreciate your recognition of our work. It is encouraging to know that you found our study to be original, relevant, and important. We are grateful for your thoughtful review and the time you have dedicated to evaluating our manuscript. Your comments have been extremely valuable in helping us strengthen the manuscript, and we are glad that our research contributes to the broader understanding of this important topic.
Comment 2: Your paper it is very well summarised in the abstract and your key words are pertinent for the paper. Moreover, you have a very important sample and the fact that you analyse the data with PROCESS 3.0 and SPSS 21 that give more pertinence to the study. You described very well the research design, your present very well your questions and your hypothesis and you use some figures to explain better your ideas. Here I would like to ad, that you numbered question 3 as question 2, and I think you could change here. At the same time you precised very well your contribution to the scholarship.
Response 2:
Thank you for your thoughtful and constructive feedback. We greatly appreciate your positive comments regarding the abstract, key words, and the overall pertinence of our study. We are also pleased that you found our use of PROCESS 3.0 and SPSS 21 appropriate for analyzing the data and enhancing the study's rigor.
We appreciate your comment about the research design and the clarity with which we presented our research questions and hypotheses, as well as the use of figures to help illustrate our ideas. We are glad these aspects of the manuscript were effective in communicating the research methodology and findings.
Regarding your observation about the numbering of the questions, we acknowledge the issue where Question 3 was incorrectly labeled as Question 2. We have corrected this oversight in the revised manuscript and appropriately renumbered the questions to ensure consistency and clarity throughout the text.
Once again, thank you for your valuable feedback. We are pleased that our contribution to the scholarship is well-received, and we will continue to refine the manuscript in line with your suggestions.
Comment 3: Overall, I firmly believe that your manuscript is rigorous with a good methodology and originality, but I would give you some suggestions that could help you to improve a little bit. By example, your conclusions paragraph it is very short and you could cut the paragraph from the line 317 (which it is in discussion) and paste it into conclusion paragraph. Moreover, I would encourage you to use a figure to present your results too. In that way the results will be much clear to follow.
Response 3:
Thank you for your thoughtful feedback and for your positive evaluation of the rigor, methodology, and originality of our manuscript. We appreciate your suggestions for further improving the paper.
In response to your suggestion about the conclusion paragraph, we agree that it would benefit from more depth. As you recommended, we have moved the paragraph from line 317 (in the discussion section) and placed it into the conclusion paragraph. This addition now ensures that the conclusion more effectively summarizes the key findings of our study and emphasizes their implications.
Furthermore, we appreciate your recommendation to include a figure to present our results more clearly. We have now added a figure to illustrate the key findings, which we believe will enhance the clarity of the results and make them easier for readers to follow.
Comment 4; Last but not least, I suggest you to detail the paragraphs in which you talk about the collectivism with some citations form the works of Hofstede, the most important author in the field of cultural dimension and collectivism. Hosftede’s work will help you to get the most out of the data you have.
Response 4:
Thank you for your valuable suggestion to further detail the paragraphs discussing collectivism. We fully appreciate your recommendation to incorporate Hofstede's work to strengthen our discussion on this cultural dimension.
In response to your comment, we have revisited the relevant paragraphs and integrated citations from Hofstede's cultural dimensions theory, particularly focusing on his work related to collectivism. This addition helps contextualize our findings within the broader framework of cultural dimensions, providing a stronger theoretical foundation for the role of self-construal and self-concept clarity (SCC) in understanding value conflicts among university students.
Specifically, we have referenced Hofstede’s theory to explain why individuals from cultures emphasizing collectivism, such as China, may experience less frequent or less intense value conflicts. As Hofstede's work suggests, collectivist cultures prioritize group harmony, which might allow individuals to reconcile conflicting values more easily than in individualistic cultures, where personal values may be in sharper conflict.
Round 2
Reviewer 1 Report
Comments and Suggestions for Authors
I am pleased with the comprehensive revisions you have made in response to my comments. The manuscript now presents a clearer structure, enhanced methodological transparency, and stronger theoretical foundations, particularly with the integration of recent literature on psychological resilience and self-concept clarity. Your attention to detail in reporting statistical analyses, including the addition of a priori power analysis, Cronbach’s alpha values, and detailed mediation and moderation results, significantly strengthens the rigor of your study. I also appreciate the expanded Discussion section, which provides valuable insights into the practical implications of your findings for mental health interventions among university students. Thank you for your diligent efforts to improve the clarity and overall quality of this important work.